# Noise-Adaptive State Estimators with Change-Point Detection

**DOI:** 10.3390/s24144585

**Published:** 2024-07-15

**Authors:** Xiaolei Hou, Shijie Zhao, Jinjie Hu, Hua Lan

**Affiliations:** College of Automation, Northwestern Polytechnical University, Xi’an 710129, China; hou.xiaolei@nwpu.edu.cn (X.H.); zhao.shi.jie@mail.nwpu.edu.cn (S.Z.); 2021262547@mail.nwpu.edu.cn (J.H.)

**Keywords:** adaptive state estimation, variational inference, change-point detection, maneuvering-target tracking

## Abstract

Aiming at tracking sharply maneuvering targets, this paper develops novel variational adaptive state estimators for joint target state and process noise parameter estimation for a class of linear state-space models with abruptly changing parameters. By combining variational inference with change-point detection in an online Bayesian fashion, two adaptive estimators—a change-point-based adaptive Kalman filter (CPAKF) and a change-point-based adaptive Kalman smoother (CPAKS)—are proposed in a recursive detection and estimation process. In each iteration, the run-length probability of the current maneuver mode is first calculated, and then the joint posterior of the target state and process noise parameter conditioned on the run length is approximated by variational inference. Compared with existing variational noise-adaptive Kalman filters, the proposed methods are robust to initial iterative value settings, improving their capability of tracking sharply maneuvering targets. Meanwhile, the change-point detection divides the non-stationary time sequence into several stationary segments, allowing for an adaptive sliding length in the CPAKS method. The tracking performance of the proposed methods is investigated using both synthetic and real-world datasets of maneuvering targets.

## 1. Introduction

State estimation of dynamical systems from noisy observations in real time is one of the most fundamental tasks in localization, tracking, and navigation [1]. The Kalman filter (KF) is an optimal state estimator for a linear Gaussian state-space model requiring known noise statistics [2]. However, in many practical situations, the statistical noise covariances are partially unknown and may abruptly change [3]. The underlying motivating application is maneuvering-target tracking [4], wherein some real-world targets, including ground vehicles, aircraft, and ballistic missiles, are capable of making very sharp, evasive maneuvers to escape from the radar’s tracking and locking. The parameters, such as the process noise covariance accounting for unexpected maneuverability, are unknown and abruptly change between different motion mode segments.

The classical solution to state estimation problems with uncertain parameters is adaptive estimators [5], which perform statistics on the model parameters or noise, as well as the dynamic state, simultaneously. Noise-adaptive estimators can be broadly divided into four categories, including Bayesian inference, maximum likelihood estimation, covariance-matching, and correlation methods [3]. Bayesian inference is the most general approach, while the other methods are often interpreted as approximations of Bayesian inference [6]. In the Bayesian inference approach, noise-adaptive estimators are required to calculate the intractable joint posterior probability density function (PDF) of the dynamic state and unknown parameters. There are three primary methods for approximating adaptive estimators, including multiple model (MM) methods, sequence Monte Carlo (SMC) methods, and variational Bayesian (VB) methods.

MM methods [7] regard the underlying dynamics as switching systems among a finite number of models, representing different noise levels or system structures. There exist both continuous noise uncertainties and discrete model uncertainties in switching systems. MM methods carry out state estimation and model selection recursively, which can be divided into static and dynamic MM estimators depending on whether the model is switched. By assuming that the model switching process is a Markov process, the well-known interacting MM (IMM) [8,9] estimator achieves a trade-off between computation and accuracy, which has proven to be promising for tracking highly maneuvering targets [1]. The applicability of IMM estimators depends on the completeness of the model sets and suffers from the curse of dimensionality. Some extended methods can tackle these issues [10].

SMC methods [11] approximate the intractable joint PDF by propagating a set of random particles, drawing from the tractable proposal distribution. For adaptive state estimation with static parameters, one solution is based on artificial parameter evolution, which perturbs particles by adding artificial noise to avoid over-diffuse approximations [12]. An alternative approach is based on particle learning [13], which marginalizes the static parameters out of the posterior distribution and can be implemented in an online fashion by constructing sufficient statistics [14]. Nemeth et al. [15] extended the work of [12] to time-varying parameters by combining SMC approaches with change-point models. Arnold [16] presented two SMC-based adaptive estimators with time-varying parameters using the concept of artificial parameter evolution. SMC methods provide a flexible and accurate Bayesian inference for adaptive estimation but are limited to small-scale state estimation problems due to their demand for massive computational power. Meanwhile, the performance of SMC methods depends on the choice of the proposal distribution. Improper design of the proposal distribution often leads to poor approximation, especially for high-dimensional estimation problems.

VB methods [17] approximate the intractable joint PDF through optimization. The VB-based adaptive state estimator has received significant attention due to its computational efficiency compared to SMC methods. Most existing VB methods address the adaptive state estimation for a linear state-space model with unknown measurement noise covariance (MNC). By modeling the conjugate prior distribution of the state and MNC as a Gaussian inverse-Wishart distribution, the joint posterior PDFs are approximated using factorized free-form (mean-field approximation) and updated via the coordinate ascent method. Särkkä and Nummenmaa [6] presented the first VB-based adaptive Kalman filtering with unknown MNC. The extension to unknown process noise covariance (PNC) is not straightforward because the joint prior of the state and PNC is non-conjugate. Huang et al. [18] extended the work of [6] to both unknown MNC and PNC by regarding the state-predicted covariance as an inverse-Wishart distribution. Ma et al. [19] constructed the conjugate prior distribution of the state and PNC by inducing auxiliary latent variables. Ardeshiri et al. proposed an adaptive smoother with unknown PNC and MNC. Xu et al. [20] proposed adaptive fixed-lag smoothing with unknown MNC. Ma et al. [19] proposed VB-based joint state estimation and model identity for multiple model systems. Zhu et al. [21] proposed an outlier-robust variational Kalman filter by leveraging Student-t noise modeling. Yu and Meng [22] proposed robust Kalman filters with multiplicative noise modeling. Xia et al. [23] proposed an adaptive variational Kalman filter with unknown MNC to solve the calibration problem. Zhu et al. [24,25] proposed variational Kalman filters with unknown, time-varying, and non-stationary heavy-tailed process and measurement noises. Huang et al. [26] proposed an adaptive Kalman filter with a Gaussian inverse-Wishart mixture distribution for unknown MNC. For nonlinear state-space models, it is intractable to directly optimize the objective with the coordinate ascent method. The basic idea is to approximate the intractable expectation of nonlinear expressions, such as adaptive Metropolis sampling [27] and the cubature integration rule [28]. An alternative approach is to employ stochastic gradient methods [29]. Lan et al. [30] proposed a nonlinear adaptive Kalman filter with unknown PNC based on stochastic search VB, achieving high estimation accuracy but suffering from slow iteration convergence.

As stated in [26], existing VB-based adaptive Kalman filters (AKF) are quite sensitive to the initial value setting of PNC. This is mainly because the Kullback–Leibler (KL) divergence is generally a nonconvex objective function, and the coordinate ascent method only guarantees convergence to a local optimum, which can be sensitive to initialization. In the domain of maneuvering-target tracking, the dimension of latent variables (target state, PNC, MNC) is generally larger than the dimension of measurements, making it easy for variational iterations to converge to local minima. As a result, initialization issues hinder the application of existing adaptive Kalman filters to sharply maneuvering-target-tracking problems, where PNC may change abruptly and accurate prior information is unavailable. In order to deal with tracking sharply maneuvering targets, an adaptive initialization strategy should be addressed.

Motivated by the challenges of tracking sharply maneuvering targets, this paper develops novel variational adaptive state estimators for joint target state and process noise parameter estimation for a class of linear state-space models with abruptly changing parameters. By combining variational inference with change-point detection in an online Bayesian fashion, two adaptive estimators—a change-point-based adaptive Kalman filter (CPAKF) and a change-point-based adaptive Kalman smoother (CPAKS)—are proposed in a recursive detection and estimation process. In each step, the run-length probability of the current maneuver mode is first calculated, and then the joint posterior of the target state and process noise parameter conditioned on the run length is approximated by variational inference. Compared with existing variational noise-adaptive Kalman filters, the proposed methods are robust to initial iterative value settings, improving their ability to track sharply maneuvering targets. Meanwhile, the change-point detection divides the non-stationary time sequence into several stationary segments, allowing for an adaptive sliding length in the CPAKS method. Finally, the superior tracking performance of the proposed methods is verified using both synthetic and real-world datasets of maneuvering-target tracking.

The remainder of this paper is organized as follows. Section 2 describes the problem formulation of adaptive state estimation with unknown process noise covariance. Section 3 presents the proposed VB-based adaptive state estimation with change-point detection. Section 5 and Section 6 provide performance comparisons using simulated and real data, respectively. Finally, Section 7 concludes this paper.

## 2. Problem Description

Maneuvering-target tracking can be characterized by the following discrete-time state-space model
(1)$$\begin{array}{cc} x_{k} & =F_{k}x_{k-1}+v_{k} \end{array}$$
(2)$$\begin{array}{cc} y_{k} & =H_{k}x_{k}+w_{k} \end{array}$$
where $x_{k}\in R^{n_{x}}$ is the target kinematic state of dimension $n_{x}$ and $y_{k}\in R^{n_{y}}$ is the sensor measurement of dimension $n_{y}$. The state transition matrix $F_{k}$ and measurement matrix $H_{k}$ are assumed to be known. The process noise vector $v_{k}\in R^{n_{x}}$ and measurement noise vector $w_{k}\in R^{n_{y}}$ are the Gaussian distribution with zero mean and the corresponding covariance matrices $Q_{k}$ and $R_{k}$, respectively. Assume that the initial state satisfies $x_{0}∼N\left({\hat{x}}_{0|0},P_{0|0}\right)$ and the random variables $x_{0}$, $v_{k}$, and $w_{k}$ are independent of each other.

In modern target tracking, non-cooperative targets are capable of making very sharp and evasive maneuvers to avoid tracking and locking on. Assume that a hostile aircraft initially travels at a constant velocity (CV) of 200 m/s for 33 s, then enters a constant turn (CT) of 10 deg/s for 33 s, and the target accelerates in a straight line at 3 $m/s^{2}$. As shown in the upper figure in Figure 1, when using the nominal CV model to represent the target motion, the magnitudes of modeling errors in the x-axis resulting from target maneuvers vary over time and change significantly when the model is switched. As shown in the lower figure in Figure 1, the change points can divide the target flight into several non-overlapping motion segments, known as the *run length*, which refers to the length of flight time since the last change point [31], and the model parameters during each run-length segment are assumed to be constant or slow-varying. The objective of this paper is to perform adaptive state estimation by continuously adjusting the process noise level.

**Definition 1** ([31])**.** *Define the discrete random variable $r_{k}\in \left\{1,\ldots ,k\right\}$ as the run length at time k. At each time k, the run length $r_{k}$ has only two outcomes: it either continues to grow $r_{k}=r_{k-1}+1$ if no change point occurs or drops to $r_{k}=1$ when a change point occurs.*

**Remark 1.** 
*The run-length variable $r_{k}$ is used to describe changes in the kinematic model or process noise level. Taking Figure 1 as an example, during the CV motion segment from 0 to 39 s, the run length continues to grow ($r_{k}=r_{k-1}+1$) with $r_{1}=1$. At time k=40, the CV motion becomes CT motion, the run-length variable at k=40 is $r_{40}=1$, and the run-length variable begins to grow again ($r_{k}=r_{k-1}+1$) with the initial value $r_{40}=1$. Similarly, the run-length variable begins to grow again ($r_{k}=r_{k-1}+1$) with the initial value $r_{70}=1$ during the CA motion segment.*


For noncooperative maneuvering-target tracking, the process noise covariance $Q_{k}$, which accounts for motion model uncertainty, is commonly unknown and time-varying, and it should be estimated along with the target state $x_{k}$. The VB-based AKF approximates the intractable joint posterior PDFs $p(x_{k},Q_{k}|y_{1:k})$ through optimization, whereas the posterior of each latent variable ($x_{k}$ and $Q_{k}$) is updated iteratively via the coordinate ascent method. Some variational Bayesian adaptive Kalman filters (VBAKF) have been proposed in [18,26], where the unknown process noise covariance is assumed to vary slowly. However, these methods are quite sensitive to the initial values due to the local optimality of VB, limiting their ability to track sharply maneuvering targets [26].

To tackle the iterative initialization problem, we introduce the run length $r_{k}$ as an auxiliary latent variable to enhance the performance of AKF. The run length $r_{k}$ divides the sequence of model parameters $Q_{k}$ into non-overlapping segments. Within each segment, the parameters $Q_{k}$ are dependent. However, across different segments, the parameters $Q_{k}$ are independent of each other. Therefore, the initial value of $Q_{k}$ at each iteration is determined by the specific run length $r_{k}$.

The objective of AKF with the presence of unknown $Q_{k}$ and $r_{k}$ is to compute the posterior PDF $p(x_{k},Q_{k},r_{k}|y_{1:k})$. Formally, the well-known recursive Bayesian optimal filtering consists of the following prediction-update cycle after starting from the prior PDF $p(x_{0},Q_{0},r_{0})$:*Prediction*: Following the Chapman–Kolmogorov equation, the predictive PDF of joint latent variables is
(3)$$\begin{array}{c} p\left(x_{k},Q_{k},r_{k}|y_{1:k-1}\right)=\sum_{r_{k-1}}\int p\left(x_{k}|x_{k-1},Q_{k}\right)\\ \times p\left(Q_{k}|Q_{k-1},r_{k}\right)p\left(r_{k}|r_{k-1}\right)\\ \times p\left(x_{k-1},Q_{k-1},r_{k-1}|y_{1:k-1}\right)dx_{k-1}dQ_{k-1}. \end{array}$$*Update*: Given the measurement $y_{k}$, the posterior PDF is updated using Bayes’ rule:
(4)$$\begin{array}{c} p\left(x_{k},Q_{k},r_{k}|y_{1:k}\right)\propto p\left(y_{k}|x_{k},Q_{k},r_{k}\right)\\ \times p(x_{k},Q_{k},r_{k}|y_{1:k-1}) \end{array}$$

Due to the intractable integrations and summarization involved in the prediction-update cycle, the general Bayesian filtering solution is not analytically tractable. We effectively solve the above recursive equations using VB, where the intractable posterior PDF $p(x_{k},Q_{k},r_{k}|y_{1:k})$ is approximated by a variational distribution $q(x_{k},Q_{k},r_{k})$ by minimizing the Kullback–Leibler (KL) divergence. Minimizing the KL divergence to zero guarantees that the variational distribution matches the exact posterior.

## 3. Adaptive Kalman Filter with Change-Point Detection

Recall that recursive Bayesian optimal filtering consists of initialization and the recursive prediction-update cycle. The initialization step involves modeling the prior distribution of latent variables at the previous time step. Given the measurements $y_{1},\cdots ,y_{k-1}$, the joint conditional PDF for $x_{k-1}$, $Q_{k-1}$, and $r_{k-1}$ is assumed to be factorized as
(5)$$\begin{array}{c} p\left(x_{k-1},Q_{k-1},r_{k-1}|y_{1:k-1}\right)=N\left(x_{k-1}|{\hat{x}}_{k-1|k-1}^{r},P_{k-1|k-1}^{r}\right)\\ \times \mathrm{IW}\left(Q_{k}|{\hat{u}}_{k-1|k-1}^{r},U_{k-1|k-1}^{r}\right)p\left(r_{k-1}|y_{1:k-1}\right) \end{array}$$
where the priors of $x_{k-1}$ and $Q_{k-1}$ are, respectively, a Gaussian distribution and an inverse-Wishart distribution with parameters associated with the run length $r_{k-1}$. The notations N(z|d,D) and IW(Z|a,A) denote the Gaussian and IW distributions, with the PDF functions given by
(6)$$\begin{array}{c} N\left(z|d,D\right)=\frac{\exp\left[-0.5{\left(z-d\right)}^{⊤}D^{-1}\left(z-d\right)\right]}{\sqrt{{\left(2\pi \right)}^{n}\left|D\right|}} \end{array}$$
(7)$$\begin{array}{cc} \mathrm{IW}(Z|a,A)= & \frac{{\left|A\right|}^{0.5a}}{2^{0.5na}\Gamma_{n}\left(0.5a\right)}{\left|Z\right|}^{-0.5(a+n+1)}\\ \; & \times \exp\left[-0.5\mathrm{Tr}\left(AZ^{-1}\right)\right] \end{array}$$
where $z\in R^{n\times 1}$ is a Gaussian random variable with mean d and covariance D, and $Z\in R^{n\times n}$ is an IW random variable with *a* and A being the degrees of freedom and positive-definite scale matrix, respectively.

### 3.1. Time Prediction Step

According to the prediction equation in (Equation 3), the joint predictive PDF of latent variables can be factorized as
(8)$$\begin{array}{c} p\left(x_{k},Q_{k},r_{k}|y_{1:k-1}\right)=p\left(x_{k}|Q_{k},r_{k},y_{1:k-1}\right)\\ \times p\left(Q_{k}|r_{k},y_{1:k-1}\right)p\left(r_{k}|y_{1:k-1}\right) \end{array}$$

The state prediction $x_{k}$ is a Gaussian distribution
(9)$$\begin{array}{c} p\left(x_{k}|Q_{k},r_{k},y_{1:k-1}\right)=N\left(x_{k}|{\hat{x}}_{k|k-1}^{r},P_{k|k-1}^{r}\right) \end{array}$$
with mean ${\hat{x}}_{k|k-1}^{r}$ and covariance $P_{k|k-1}^{r}$ given by
(10)$$\begin{array}{cc} {\hat{x}}_{k|k-1}^{r}= & F_{k}{\hat{x}}_{k-1|k-1},\\ P_{k|k-1}^{r}= & F_{k}P_{k-1|k-1}F_{k}+{\hat{Q}}_{k}^{r}, \end{array}$$
where ${\hat{Q}}_{k}^{r}$ is the expectation of $Q_{k}$, defined by
(11)$$\begin{array}{c} {\hat{Q}}_{k}^{r}≜\;E_{p(Q_{k}|r_{k},y_{1:k-1})}\left[Q_{k}\right]=\;\frac{U_{k|k-1}^{r}}{{\hat{u}}_{k|k-1}^{r}-n_{x}-1} \end{array}$$

The PNC prediction $Q_{k}$ is an inverse-Wishart distribution
(12)$$\begin{array}{c} p\left(Q_{k}|r_{k},y_{1:k-1}\right)=\mathrm{IW}\left(Q_{k}|{\hat{u}}_{k|k-1}^{r},U_{k|k-1}^{r}\right) \end{array}$$
with the parameters conditional on $r_{k}$ given by
(13)$$\begin{array}{c} {\hat{u}}_{k|k-1}^{r}=\left\{\begin{array}{cc} \beta \left({\hat{u}}_{k-1|k-1}^{r}-n_{x}-1\right)+n_{x}+1, & r_{k}=r_{k-1}+1\\ \beta \left(u_{0}-n_{x}-1\right)+n_{x}+1, & r_{k}=1 \end{array}\right. \end{array}$$
and
(14)$$\begin{array}{c} U_{k|k-1}^{r}=\left\{\begin{array}{cc} \beta U_{k-1|k-1}^{r}, & r_{k}=r_{k-1}+1\\ \beta U_{0}, & r_{k}=1 \end{array}\right. \end{array}$$
where β∈(0,1] denotes the forgetting factor of $Q_{k}$. The predefined parameters $u_{0}$ and $U_{0}$ are the initial values of PNC $Q_{k}$ when the change point occurs at time *k*.

**Remark 2.** 
*Note that the detailed dynamical model of NPC, denoted as $p(Q_{k}|Q_{k-1})$, is unavailable. Most VB-based AKF methods employ heuristic dynamics that simply propagate their previous approximate posteriors. This heuristic is stable during VB iteration when $Q_{k}$ is stationary or slow-varying. However, it is not applicable for non-stationary or fast-varying $Q_{k}$ in sharply maneuvering-target tracking.*


The predictive PDF of run length $r_{k}$ can be factorized as
(15)$$\begin{array}{c} p\left(r_{k}|y_{1:k-1}\right)=\sum_{r_{k-1}}p\left(r_{k}|r_{k-1}\right)p\left(r_{k-1}|y_{1:k-1}\right) \end{array}$$
where $p(r_{k}|r_{k-1})$ is the transition probability of run length $r_{k}$. According to the definition of the run length, the transition probability has only two outcomes: the run length either continues to grow or a change point occurs. We have
(16)$$p\left(r_{k}|r_{k-1}\right)=\left\{\begin{array}{cc} 1-G(r_{k-1}+1), & r_{k}=r_{k-1}+1\\ G(r_{k-1}+1), & r_{k}=1 \end{array}\right.$$
where G(·) denotes the hazard function representing the prior distribution of the change point. To incorporate flexibility, various complex hazard function models can be utilized. For simplicity, we choose the geometric distribution, and the hazard function is given by [31]
(17)$$G\left(r_{k}\right)=\frac{1}{\lambda }$$
where λ is the timescale parameter of the change point.

Substituting (Equation 16) and (Equation 17) into (Equation 15) yields
(18)$$p\left(r_{k}|y_{1:k-1}\right)=\left\{\begin{array}{c} \frac{1-\lambda }{\lambda }p\left(r_{k-1}|y_{1:k-1}\right),\;r_{k}=r_{k-1}+1\\ \frac{1}{\lambda }\sum_{r_{k-1}}p\left(r_{k-1}|y_{1:k-1}\right),\;r_{k}=1 \end{array}\right.$$

### 3.2. Posterior Update Step

In the posterior update step, the joint posterior PDF of latent variables $x_{k}$, $Q_{k}$, and $r_{k}$ can be factorized as
(19)$$\begin{array}{c} p\left(x_{k},Q_{k},r_{k}|y_{1:k}\right)=p\left(r_{k}|y_{1:k}\right)p\left(x_{k},Q_{k}|r_{k},y_{1:k}\right) \end{array}$$

The posterior PDF of the run length $r_{k}$ can be derived via the Bayesian theorem:(20)$$\begin{array}{cc} p(r_{k}|y_{1:k})= & \frac{p\left(r_{k}|y_{1:k-1}\right)p\left(y_{k}|r_{k},y_{1:k-1}\right)p\left(y_{k-1}\right)}{p(y_{1:k})}\\ \propto & p\left(r_{k}|y_{1:k-1}\right)\Lambda_{k}^{r} \end{array}$$
where $\Lambda_{k}^{r}≜p\left(y_{k}|r_{k},y_{1:k-1}\right)$ is given by
(21)$$\begin{array}{cc} \Lambda_{k}^{r}= & \int p\left(y_{k}|x_{k},R_{k}\right)p\left(x_{k}|r_{k},Q_{k},y_{1:k-1}\right)dx_{k}\\ = & N(y_{k}|H_{k}{\hat{x}}_{k|k-1}^{r},H_{k}P_{k|k-1}^{r}H_{k}^{⊤}+R_{k}) \end{array}$$

Conditioned on the run length $r_{k}$, the state $x_{k}$ and PNC $Q_{k}$ are coupled through the likelihood $p(y_{k}|x_{k},Q_{k})$, making the exact conditional posterior $p(x_{k},Q_{k}|r_{k},y_{1:k})$ devoid of a tractable analytic form. One must resort to the approximate Bayesian inference technique. The standard VB method is employed to approximate the exact conditional posterior with a tractable variational distribution $q(x_{k},Q_{k}|r_{k})$, which takes the form of factorized free factors as follows
(22)$$\begin{array}{cc} q(x_{k},Q_{k}|r_{k})\approx & q\left(x_{k}|r_{k}\right)q\left(Q_{k}|r_{k}\right)\\ = & N\left(x_{k}|{\hat{x}}_{k|k}^{r},P_{k|k}^{r}\right)\mathrm{IW}\left(Q_{k}|{\hat{u}}_{k|k}^{r},U_{k}^{r}\right) \end{array}$$

The variational hyperparameters $\lambda_{k}≜\left\{\lambda_{x}^{r},\lambda_{Q}^{r}\right\}$ can be obtained by minimizing the KL divergence between the variational distribution and the exact posterior, where $\lambda_{x}^{r}=\left\{{\hat{x}}_{k|k}^{r},P_{k|k}^{r}\right\}$ and $\lambda_{Q}^{r}=\left\{{\hat{u}}_{k|k}^{r},U_{k|k}^{r}\right\}$. Minimizing the KL divergence is equivalent to maximizing the evidence lower bound (ELBO) $B(\lambda_{k})$, defined as
(23)$$\begin{array}{cc} B\left(\lambda_{k}\right)=E_{q(x_{k},Q_{k}|r_{k})}[ & \log p(x_{k},Q_{k},y_{k}|r_{k},y_{1:k-1})\\ \; & -\log q(x_{k},Q_{k}|r_{k})] \end{array}$$
where the joint PDF $J_{k}≜p\left(x_{k},Q_{k},y_{k}|r_{k},y_{1:k-1}\right)$ can be formulated as
(24)$$\begin{array}{cc} J_{k}= & p\left(y_{k}|x_{k},R_{k}\right)p\left(x_{k}|r_{k},Q_{k},y_{1:k-1}\right)\\ \; & \times \log p(Q_{k}|r_{k},y_{1:k-1})\\ = & N\left(y_{k}|H_{k}x_{k},R_{k}\right)N\left(x_{k}|{\hat{x}}_{k|k-1}^{r},P_{k|k-1}^{r}\right)\\ \; & \times \mathrm{IW}(Q_{k}|{\hat{u}}_{k|k-1}^{r},U_{k|k-1}^{r}) \end{array}$$

Substituting (Equation 22) and (Equation 24) into (Equation 23), the ELBO $B(\lambda_{k})$ can be rewritten as
(25)$$\begin{array}{cc} B(\lambda_{k})= & E_{q(x_{k}|r_{k})}\left[\log N\left(y_{k}|H_{k}x_{k},R_{k}\right)\right]\\ \; & +E_{q(x_{k},Q_{k}|r_{k})}\left[\log N\left(x_{k}|{\hat{x}}_{k|k-1}^{r},P_{k|k-1}^{r}\right)\right]\\ \; & -E_{q(x_{k}|r_{k})}\left[\log N\left(x_{k}|{\hat{x}}_{k|k}^{r},P_{k|k}^{r}\right)\right]\\ \; & +E_{q(Q_{k}|r_{k})}\left[\log\mathrm{IW}\left(Q_{k}|{\hat{u}}_{k|k-1}^{r},U_{k|k-1}^{r}\right)\right]\\ \; & -E_{q(Q_{k}|r_{k})}\left[\log\mathrm{IW}\left(Q_{k}|{\hat{u}}_{k|k}^{r},U_{k|k}^{r}\right)\right] \end{array}$$

Then, the optimal hyperparameters of $x_{k}$ and $Q_{k}$ can be obtained as follows:(26)$$\begin{array}{c} \left\{{\left(\lambda_{x}^{r}\right)}^{*},{\left(\lambda_{Q}^{r}\right)}^{*}\right\}=\underset{\lambda_{x}^{r},\lambda_{Q}^{r}}{\mathrm{argmax}}\;B\left(\lambda_{k}\right) \end{array}$$

**Remark 3.** *By maximizing the ELBO in *(Equation 26)*, the optimal hyperparameters of $x_{k}$ and $Q_{k}$ are derived by setting the gradients of $B(\lambda_{k})$ with respect to the corresponding hyperparameters to zero [32]. The expectation parameters of the posterior distributions are used in the following derivation. For $q(x_{k}|r_{k})$ and $q(Q_{k}|r_{k})$, the expectation parameters are $\left\{E\left[x_{k}\right],E\left[x_{k}x_{k}^{⊤}\right]\right\}$ and $\{E\left[Q_{k}^{-1}\right],E[\log\left|Q_{k}|]\right\}$, respectively. For simplicity, the expectation $E_{q(\cdot )}$ is written as E.*

Rewrite the EBLO $B(\lambda_{k})$ as the function of $x_{k}$, and omit the rest independent terms, denoted as $B_{x}\left(\lambda_{x}^{r}\right)$. One has
(27)$$\begin{array}{cc} B_{x}\left(\lambda_{x}^{r}\right)= & -\frac{1}{2}E\left\{\mathrm{tr}[R_{k}^{-1}\left(y_{k}-H_{k}x_{k}\right){\left(\cdot \right)}^{⊤}]\right\}\\ \; & -\frac{1}{2}E\left\{\mathrm{tr}[{\left(P_{k|k-1}^{r}\right)}^{-1}\left(x_{k}-{\hat{x}}_{k|k-1}^{r}\right){\left(\cdot \right)}^{⊤}]\right\}\\ \; & +\frac{1}{2}E\left\{\mathrm{tr}[{\left(P_{k|k}^{r}\right)}^{-1}\left(x_{k}-{\hat{x}}_{k|k}^{r}\right){\left(\cdot \right)}^{⊤}]\right\} \end{array}$$
where tr(A) denotes the trace of matrix A and $A{\left(\cdot \right)}^{⊤}≜{\mathrm{AA}}^{⊤}$.

Simplify the ELBO $B_{x}\left(\lambda_{x}^{r}\right)$ as a formulation with respect to its expectation parameters $\left\{E\left[x_{k}\right],E\left[x_{k}x_{k}^{⊤}\right]\right\}$. The remaining independent terms can be ignored since their gradients with respect to the expectation parameters are zero. One has
(28)$$\begin{array}{cc} B_{x}\left(\lambda_{x}^{r}\right)= & -\frac{1}{2}E\left\{\mathrm{Tr}[H_{k}^{⊤}R_{k}^{-1}\left(-2y_{k}x_{k}^{⊤}+H_{k}x_{k}x_{k}^{⊤}\right)]\right\}\\ \; & -\frac{1}{2}E\left\{\mathrm{Tr}\left[{\left(P_{k|k-1}^{r}\right)}^{-1}\left(-2{\hat{x}}_{k|k-1}^{r}x_{k}^{⊤}+x_{k}x_{k}^{⊤}\right)\right]\right\}\\ \; & +\frac{1}{2}E\left\{\mathrm{Tr}\left[{\left(P_{k|k}^{r}\right)}^{-1}\left(-2{\hat{x}}_{k|k}^{r}x_{k}^{⊤}+x_{k}x_{k}^{⊤}\right)\right]\right\}\\ = & \mathrm{Tr}\left[\left(H_{k}^{⊤}R_{k}^{-1}y_{k}+E\left[{\left(P_{k|k-1}^{r}\right)}^{-1}\right]{\hat{x}}_{k|k-1}^{r}-{\left(P_{k|k}^{r}\right)}^{-1}{\hat{x}}_{k|k}^{r}\right)E{\left[x_{k}\right]}^{⊤}\right]\;-\\ \; & \frac{1}{2}\mathrm{Tr}\left[\left(H_{k}^{⊤}R_{k}^{-1}H_{k}+E\left[{\left(P_{k|k-1}^{r}\right)}^{-1}\right]-{\left(P_{k|k}^{r}\right)}^{-1}\right)E\left[x_{k}x_{k}^{⊤}\right]\right] \end{array}$$

Let the gradients with respect to the expectation parameters $\left\{E\left[x_{k}\right],E\left[x_{k}x_{k}^{⊤}\right]\right\}$ be equal to zero. The optimal hyperparameters are obtained as follows:(29)$$\begin{array}{cc} {\left(P_{k|k}^{r}\right)}^{-1}= & E\left[{\left(P_{k|k-1}^{r}\right)}^{-1}\right]+H_{k}^{⊤}R_{k}^{-1}H_{k}\\ {\left(P_{k|k}^{r}\right)}^{-1}{\hat{x}}_{k|k}^{r}= & E\left[{\left(P_{k|k-1}^{r}\right)}^{-1}\right]{\hat{x}}_{k|k-1}^{r}+H_{k}^{⊤}R_{k}^{-1}y_{k} \end{array}$$

**Remark 4.** 
*Since the exact solution of $E\left[{\left(P_{k|k-1}^{r}\right)}^{-1}\right]=E_{q(Q_{k}|r_{k})}{\left[F_{k}P_{k-1|k-1}F_{k}^{⊤}+Q_{k}\right]}^{-1}$ is intractable, as an alternative approximation, we assume that $E\left[{\left(P_{k|k-1}^{r}\right)}^{-1}\right]\approx {\left[F_{k}P_{k-1|k-1}F_{k}^{⊤}+E\left(Q_{k}\right)\right]}^{-1}$.*


After obtaining the conditional distribution $q(x_{k}|r_{k})$, the approximate variational distribution $q(x_{k})$ is given by
(30)$$\begin{array}{cc} q(x_{k})= & \sum_{r_{k}}q\left(x_{k}|r_{k}\right)p\left(r_{k}|y_{1:k}\right)\\ = & N(x_{k}|{\hat{x}}_{k|k},P_{k|k}) \end{array}$$
where
(31)$$\begin{array}{cc} P_{k|k}^{-1}= & \sum_{r_{k}}p\left(r_{k}|y_{1:k}\right){\left(P_{k|k}^{r}\right)}^{-1}\\ P_{k|k}^{-1}{\hat{x}}_{k|k}= & \sum_{r_{k}}p\left(r_{k}|y_{1:k}\right){\left(P_{k|k}^{r}\right)}^{-1}{\hat{x}}_{k|k}^{r} \end{array}$$

Similarly, rewrite the EBLO $B(\lambda_{k})$ as a function of $Q_{k}$ and omit the remaining terms dependent on $Q_{k}$, denoted as $B_{Q}\left(\lambda_{Q}^{r}\right)$. Then, one has
(32)$$\begin{array}{cc} B_{Q}\left(\lambda_{Q}^{r}\right)= & E[\log N\left(x_{k}|{\hat{x}}_{k|k-1}^{r},P_{k|k-1}^{r}\right)]\\ \; & +E[\log\mathrm{IW}\left(Q_{k}|{\hat{u}}_{k|k-1}^{r},U_{k|k-1}^{r}\right)]\\ \; & -E[\log\mathrm{IW}\left(Q_{k}|{\hat{u}}_{k|k}^{r},U_{k}^{r}\right)] \end{array}.$$

It can be seen that the state $x_{k}$ and PNC $Q_{k}$ are coupled through the predictive PDF $N(x_{k}|{\hat{x}}_{k|k-1}^{r},P_{k|k-1}^{r})$. It is intractable to directly optimize the ELBO objective due to the non-conjugate problem. In order to make the computations tractable, the following assumption is made:(33)$$\begin{array}{cc} \log N(x_{k}| & {\hat{x}}_{k|k-1}^{r},P_{k|k-1}^{r})\\ = & \log E_{q(x_{k-1})}\left[N\left(x_{k}|F_{k}x_{k-1},Q_{k}\right)\right]\\ \approx & E_{q(x_{k-1})}\log\left[N\left(x_{k}|F_{k}x_{k-1},Q_{k}\right)\right] \end{array}$$

Substituting (Equation 33) into (Equation 32), one has
(34)$$\begin{array}{cc} B_{Q}\left(\lambda_{Q}^{r}\right)= & \frac{1}{2}\mathrm{tr}\left\{\left(U_{k|k}^{r}-U_{k|k-1}^{r}-A_{k}\right)E\left[Q_{k}^{-1}\right]\right\}\\ \; & +\frac{1}{2}\left({\hat{u}}_{k|k}^{r}-{\hat{u}}_{k|k-1}^{r}-1\right)E[\log|Q_{k}\left|\right] \end{array}$$

Let the gradients of $B_{Q}\left(\lambda_{Q}^{r}\right)$ with respect to the expectation parameters $\{E\left[Q_{k}^{-1}\right],E[\log|Q_{k}|]\}$ be equal to zero. The optimal hyperparameters of $Q_{k}$ can be obtained as
(35)$$\begin{array}{cc} {\hat{u}}_{k|k}^{r}= & {\hat{u}}_{k|k-1}^{r}+1\\ U_{k|k}^{r}= & U_{k|k-1}^{r}+A_{k} \end{array}$$
where the sufficient statistic $A_{k}$ can be calculated as [33]
(36)$$\begin{array}{cc} A_{k}= & E_{q(x_{k}|r_{k})}\left\{E_{q(x_{k-1})}\left[\left(x_{k}-F_{k}x_{k-1}\right){\left(\cdot \right)}^{⊤}\right]\right\}\\ = & \left({\hat{x}}_{k|k}^{r}-F_{k}{\hat{x}}_{k-1|k-1}\right){\left(\cdot \right)}^{⊤}+F_{k}P_{k-1|k-1}F_{k}^{⊤}\\ \; & +P_{k|k}^{r}-P_{k,k-1}^{⊤}F_{k}^{⊤}-F_{k}P_{k,k-1} \end{array}$$
where $P_{k,k-1}=P_{k-1|k-1}F_{k}^{⊤}{\left(P_{k|k-1}^{r}\right)}^{-1}P_{k|k}^{r}$.

The proposed AKF based on change-point detection, referred to as CPAKF, is summarized in Algorithm 1.
**Algorithm 1: CPAKF** (at time *k*).
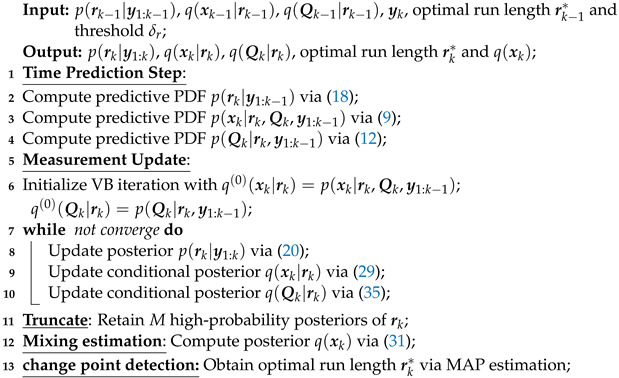


**Remark 5.** 
*CPAKF is distinct from other adaptive estimators. For instance, the IMM estimator assumes several discrete process noise levels and uses a switching rule, whereas CPAKF involves continuous process noise level adjustment through joint estimation. Compared with the existing VBAKF [18], CPAKF improves the VB initialization process by incorporating change-point detection. Unlike traditional maneuver detection methods, CPAKF relies on online Bayesian change-point detection.*


**Remark 6.** 
*CPAKF is an adaptive Kalman filter. This algorithm is suitable for sharply maneuvering-target tracking scenarios with unknown noise statistics. The adaptation means that the algorithm can perform noise parameter identification and state estimation simultaneously.*


**Remark 7.** *The convergence of the proposed CPAKF can be explained as follows. The state estimation and process noise identification of the maneuvering-target tracking are formulated as a variational optimization problem. CPAKF implements three steps to solve this optimization problem. Firstly, the posterior of the run length $p(r_{k}|y_{1:k})$ can be calculated using *(Equation 20)* via the Bayesian theorem. Secondly, the conditional posteriors $q(x_{k}|r_{k})$ and $q(Q_{k}|r_{k})$ are calculated using *(Equation 29)* and *(Equation 35)* by maximizing the ELBO *(Equation 26)*. Finally, the state posterior $q(x_{k})$ is obtained using *(Equation 31)* via the Bayesian theorem. Since the solutions obtained by the Bayesian theorem are optimal, the convergence of the CVIAKF depends on the procedure of maximizing the ELBO *(Equation 26)*. Recall that maximizing the ELBO is core to mean-field variational inference [17]. From the theoretical analysis in [34], one knows that the mean-field variational inference has good convergence, guaranteeing a linear convergence rate. Therefore, the proposed CPAKF gradually converges to a local optimum as the number of iterations increases.*

### 3.3. Implementation Details

Based on the Bayesian filtering framework, CPAKF consists of a recursive prediction-update cycle. To deal with state estimation of uncertain NPC, AKF carries out the measurement update via VB iteration. Meanwhile, change-point detection is employed to enhance the initialization of VB iteration. Some implementation details are as follows:**Reduce computation complexity:** In our algorithm, the run length $r_{k}\in \left\{1,2,\cdots ,k\right\}$ is introduced to characterize change points. The number of its probability values depends on *k*, which means the number of posterior $p(r_{k}|y_{1:k})$ will increase with *k*, resulting in an explosion of computation complexity. This problem can be mitigated by truncating less probable events. Two main truncating strategies can be employed, as suggested in [31]. One approach is to remove events where the probability of $r_{k}$ is less than a threshold $\delta_{p}$. Another method involves retaining *M* high-probability events of $r_{k}$ and normalizing their posterior probabilities. In this article, the second method is adopted to guarantee robustness across different scenarios.**Change-point detection:** In order to determine whether a change point has occurred, the maximum a posteriori (MAP) is exploited to estimate the optimal run length $r_{k}^{*}$:
(37)$$\begin{array}{c} r_{k}^{*}=\arg\max_{r_{k}}\;p\left(r_{k}|y_{1:k-1}\right) \end{array}$$
where $r_{k}^{*}$ represents the optimal run length at time *k*. According to the definition of the run length $r_{k}$, a change point is detected when $r_{k}^{*}=1$; otherwise, no change point occurs.

## 4. Adaptive Kalman Smoother Based on Change-Point Detection

In the previous section, we proposed the CPAKF algorithm. When detecting a change point, the process noise parameters can be initialized in accordance with the maneuvering scenario. To further improve estimation performance, fixed-interval adaptive Kalman smoothing based on change-point detection is proposed in this section, which can be accomplished based on CPAKF.

### 4.1. Update of the Joint Posterior

Consider the measurement set $y_{D}$ with interval D=[k−l,k], where *l* denotes the length of the interval. The objective of smoothing is to obtain the posterior estimation of smoothing of the joint latent variables set (i.e., $x_{D}^{s}$ and $Q_{D}^{s}$) over this interval.

At time k−l, the joint PDF $p(x_{k-l},Q_{k-l})$ can be defined as
(38)$$\begin{array}{cc} p(x_{k-l},Q_{k-l})= & q(x_{k-l};{\hat{x}}_{k-l|k-l},P_{k-l|k-l})\\ \; & \times \;q(Q_{k-l};{\hat{u}}_{k-l|k-l},U_{k-l|k-l}) \end{array}$$

Then, the joint posterior of $Z≜\{x_{D}^{s},Q_{D}^{s}\}$ can be obtained via the Bayesian rule:(39)$$\begin{array}{cc} p(Z|y_{D})\propto & p\left(Z,y_{D}\right)=p\left(x_{k-l},Q_{k-l}\right)\\ \; & \times p\left(y_{k}|x_{k}\right)\prod_{n=1}^{k-l}p\left(Q_{n+1}|Q_{n}\right)\\ \; & \times \prod_{n=0}^{k-l}p\left(y_{k}|x_{n},R_{n}\right)p\left(x_{n+1}|x_{n},Q_{n+1}\right) \end{array}$$

Since the analytical solution of the joint posterior is intractable, the mean-field variational inference can be employed to obtain the following approximate solution:(40)$$\begin{array}{c} p\left(Z|y_{D}\right)\approx q\left(Z\right)≜q\left(x_{D}^{s}\right)q\left(Q_{D}^{s}\right) \end{array}$$
where $q(x_{D}^{s})$ and $q(Q_{D}^{s})$ denote the approximate posterior distributions, respectively. Similar to the work in [33], we can obtain the optimization solution via the coordinate ascent method, that is: (41)$$\begin{array}{c} \log q\left(x_{D}\right)\overset{c}{=}E_{q(Q_{D})}\left[\log p\left(Z,y_{D}|y_{k-l:k-1}\right)\right] \end{array}$$(42)$$\begin{array}{c} \log q\left(Q_{D}\right)\overset{c}{=}E_{q(x_{D})}\left[\log p\left(Z,y_{D}|y_{k-l:k-1}\right)\right] \end{array}$$

For further derivation of (Equation 41), the forward filter from n=k−l to n=k can be updated as follows [33]:(43)x^n|n−1=Fnx^n−1|n−1(44)Pn|n−1=FnPn−1|n−1Fn⊤+Un|ns/u^n|ns(45)Kn=Pn|n−1Hn⊤(HnPn|n−1Hn⊤+Rn)−1(46)x^n|n=x^n|n−1+Kn(yn−Hnx^n|n−1)(47)Pn|n=Pn|n−1−KnHnPn|n−1
where $U_{n|n}^{s}$ and ${\hat{u}}_{n|n}^{s}$ denote the hyperparameters of $q(Q_{n}^{s})$.

The backward smoother from n=k−1 to n=k−l can be updated as
(48)Gn=Pn|nFn⊤Pn+1|n−1(49)x^n|ns=x^n|n+Gn(x^n+1|n+1s−x^n+1|n)(50)Pn|ns=Pn|n+Gn(Pn+1|n+1s−Pn+1|n)Gn⊤

For the derivation of $q(Q_{D}^{s})$, we can also obtain the following update of the forward filter from n=k−l to n=k:(51)u^n|n−1=β(u^n−1|n−1−nx−1)+nx+1(52)Un|n−1=βUn−1|n−1(53)u^n|n=u^n|n−1+1(54)Un|n=Un|n−1+Ans
where
(55)$$\begin{array}{cc} \; & A_{n}^{s}=\left({\hat{x}}_{n+1|n+1}^{s}-F_{n}{\hat{x}}_{n|n}^{s}\right){\left({\hat{x}}_{n+1|n+1}^{s}-F_{n}{\hat{x}}_{n|n}^{s}\right)}^{⊤}\\ \; & -F_{n}P_{n+1,n}^{⊤}-P_{n+1,n}F_{n}^{⊤}+P_{n+1|n+1}^{s}+F_{n}P_{n|n}^{s}F_{n}^{⊤} \end{array}$$
with $P_{n+1,n}=G_{n}P_{n+1|n+1}^{s}$. According to the beta-Bartlett model proposed in [35], the backward smoothing from n=k−1 to n=k−l can be derived as
(56)u^n|ns=(1−β)u^n|n+βu^n+1|n+1s(57)Un|ns=(1−β)Un|n−1+β(Un+1|n+1s)−1−1

### 4.2. Algorithm Details

The startup logic is introduced as follows. Record the end time of the last smoothing interval as $t_{c}$. According to (Equation 37), determine whether there exists a change point at time *k*. If a change point is detected, reset the smoothing interval to $D=[t_{c}+1,k]$ and start the variational smoothing algorithm. Note that if no change point is detected, define a maximum sliding-window length $l_{max}$. When $k-t_{c}\geq l_{max}$, execute the smoothing algorithm. Finally, by embedding the variational interval smoothing algorithm into the CPAKF filtering algorithm proposed in the previous section, we obtain a variational smoothing algorithm based on change points. The proposed smoothing algorithm, named CPAKS, is summarized in Algorithm 2.

**Remark 8.** *CPAKS is proposed to improve estimation accuracy by embedding the variational interval smoothing algorithm [33] into CPAKF. In the smoothing procedure, the optimization solution for CPAKS is derived from *(Equation 41)* and *(Equation 42)* via the coordinate ascent method. The coordinate ascent method is part of the fixed iteration method in mean-field variational inference [17]. According to the convergence guarantee in [34], CPAKS can converge to the local optimum as the number of iterations increases.*

**Algorithm 2: CPAKS**.

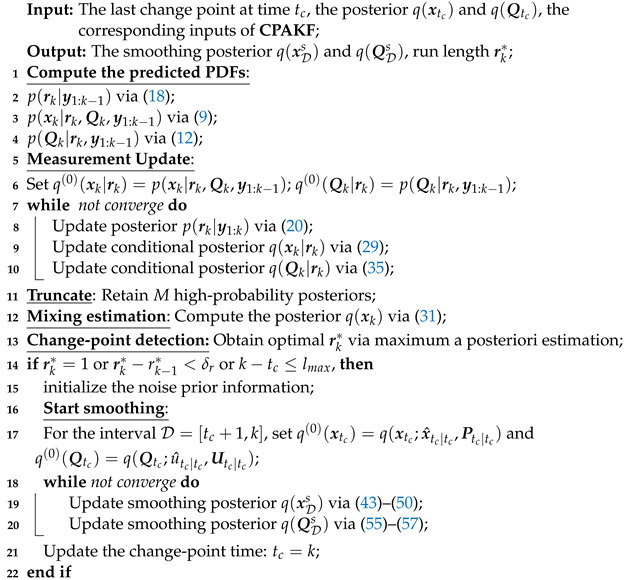



## 5. Results for Synthetic Data

To verify the effectiveness and superiority of the proposed CPAKF and CPAKS, the following simulated maneuvering-target tracking scenarios are considered.

### 5.1. Simulation Configuration

The target state is represented as $x_{k}={\left[x_{k},{\dot{x}}_{k},y_{k},{\dot{y}}_{k}\right]}^{⊤}$. The target kinematics are assumed to follow a constant-velocity kinematic model. The measurement information is obtained from a linear sensor. Then, the corresponding parameters are set as follows:(58)$$\begin{array}{c} F_{k}=I_{2}⊗\left[\begin{array}{cc} 1 & T\\ 0 & 1 \end{array}\right],\;H_{k}=\left[\begin{array}{cccc} 1 & 0 & 0 & 0\\ 0 & 0 & 1 & 0 \end{array}\right] \end{array}$$

The simulated scenarios of typical aerial maneuvering targets proposed in [36] are considered in this article, which are available in the Benchmark Trajectories for Multi-Object Tracking tool in Matlab. The measurement noise covariance $R_{k}$ is known, given by $R_{k}=\mathrm{diag}\left(10^{4},10^{4}\right)$. The process noise covariance is unknown and can be denoted as
(59)$$\begin{array}{c} Q_{k}=\delta_{k}^{2}I_{2}⊗\left[\begin{array}{cc} T^{3}/2 & T^{2}/2\\ T^{2} & T \end{array}\right] \end{array}$$
where the process noise level is defined as $\delta_{k}^{2}$ and *T* denotes the sampling period. $I_{n}$ is an n×n identity matrix. The simulated scenarios shown in Figure 2 are described as follows:**S1:** The flight trajectory of a large airplane is shown in Figure 2a. The target first flies at a constant speed of 290 m/s, and then makes a small turn at twice the gravitational acceleration. Subsequently, it begins to move at a constant speed in a straight line. Lastly, it performs a turning motion at triple the gravitational acceleration. The true moment set at which change points occur is [60,79,110,130] (second).**S2:** The second target trajectory is generated using a small aircraft. Its trajectory contains two turn movements. During this period, several accelerated movements are conducted. The set of change points in time is [31,51,101,115] (second).**S3**: The target denotes a plane moving at high velocity. The flight trajectory consists of two turning movements with accelerations four times that of gravitational acceleration. The plane slows down in the middle of the second turn. The corresponding set of times for change points is [31,40,75,91,104] (second).**S4:** In this scenario, the target and its flight trajectory are similar to those in **S3**, but the turning accelerations are 4g and 6g, respectively, where *g* denotes the gravitational acceleration. The set of times for change points is [31,36,70,81,92,142] (second).**S5:** The target is a fast-maneuvering plane. The trajectory contains three turns at a constant speed. During these periods, the target is accelerated. The set of time for change point is [31,36,63,70,118,128,133,171] (second).**S6:** The target trajectory contains four turns. After the second turn, the plane decreases its altitude and velocity and begins to conduct the third turn. Thereafter, the plane accelerates rapidly and enters into the fourth turn. The set of times for change points is [31,40,70,83,117,25,151,155] (second).

The following adaptive Kalman filtering and smoothing methods are compared with the proposed CPAKF and CPAKS. The corresponding parameters are set as follows:**IMM** [7]: The state transition $F_{k}$ is assumed to be the same as in (Equation 58). The process noise level set satisfies $\delta_{i}^{2}\in \left\{0.1\delta^{2},1\delta^{2},10\delta^{2}\right\}$. The measurement update is carried out via the standard Kalman filter. The model transition probability $P_{m}$ is set as
(60)$$\begin{array}{c} P_{m}=\left[\begin{array}{ccc} 0.9 & 0.01 & 0.09\\ 0.025 & 0.75 & 0.225\\ 0.15 & 0.35 & 0.50 \end{array}\right] \end{array}$$**VBAKF** [33]: This is an adaptive Kalman filter based on variational Bayesian methods. The initial parameters are $u_{0|0}=7$ and $U_{0|0}=3Q_{0}$, and the tuning parameter is τ=3. The decreasing factor is β=0.98, and the maximum iteration number is $I_{max}=50$.**CPAKF**: This is an adaptive Kalman filter based on change-point detection proposed by us. Most parameters are set the same as in VBAKF. In addition, the hazard function is 1/λ=0.06. The change-point parameter is $\tau_{1}=200$.**IMMS** [37]: This is an interactive multiple-model smoothing algorithm, which consists of a forward filter and a backward smoother. The smoothing interval is fixed, and its length is $l_{max}=10$. The other parameters are the same as those used in IMM.**VBAKS** [33]: This is an adaptive Kalman smoothing method. The smoothing interval is fixed, and its length is $l_{max}=10$. The other parameters are set the same as in VBAKF.**CPAKS**: This is an adaptive Kalman smoothing method based on change-point detection. The maximum smoothing interval length is $l_{max}=10$. The other parameters are similar to those of VBKAF-CP.

For each scenario, 100 Monte Carlo runs were carried out on a Legion Y9000P computer equipped with a CPU operating at 2.20 GHz. To evaluate the tracking performance of all the above methods, the Root Mean Square Error (RMSE) and Average Root Mean Square Error (ARMSE) were utilized. Additionally, to describe the abilities of CPAKF and CPAKS in change-point detection, the modified $F_{1}$ score metric proposed in [38] was used to evaluate detection accuracy. In the practical time series, the places where change points occur are influenced by some stochastic factors (e.g., process noise and measurement noise). Based on the definition of $F_{1}$, change-point detection is viewed as a classification problem of change points and invariant points. It can be defined as
(61)$$\begin{array}{c} F_{1}=\frac{2PR}{P+R} \end{array}$$
where *P* denotes precision (the ratio of correctly detected change points to the number of detected change points). *R* denotes recall (the ratio of correctly detected change points to the number of true change points). The criteria for correctly detecting change points need to be defined. Given the error margin *M* of the true change point, a detected change point is considered correct if it falls within this margin. To avoid the double-counting problem, it needs to be ensured that only one change point is detected within the error margin of each true change point. On this basis, let X denote the set of change points detected by our algorithms and T denote the set of true change points. The set of correctly detected change points in the set X is expressed as TP(T,P). For each change point τ in the set TP(T,P), there is one change point *x* in the set X satisfying |τ−x|≤M and only one *x* can match the true change point τ. Thus, precision *P* and recall *R* can be computed as
(62)$$\begin{array}{cc} P= & \frac{\left|TP(T,P)\right|}{\left|X\right|} \end{array}$$
(63)$$\begin{array}{cc} R= & \frac{\left|TP(T,P)\right|}{\left|T\right|} \end{array}$$
where |·| denotes the number of factors in the set. Note that *P* and *R* cannot be well defined if no change point is reported by our algorithms or if no true change point occurs in the scenarios. To avoid this, the point at the initial time is considered a generalized change point.

### 5.2. Simulation Results

The simulation results are shown in Figure 3. The gray dotted line indicates the moment when the target maneuvers. Among the three filtering algorithms, i.e., IMM, VBAKF, and CPAKF, the estimation accuracy of IMM in the non-maneuvering segment across all scenarios shows a significant advantage because IMM employs multiple models, allowing for better estimation through model probability weighting. Second is the CPAKF algorithm proposed by us. Its RMSE is significantly lower compared to VBAKF. This is mainly because CPAKF uses a run-length probability weighting method to obtain the maximum entropy distribution of the state estimation, which is better than the single-mode estimation used in VBAKF. In the maneuvering segments, the estimation performance of IMM is not satisfactory, especially after a change point occurs, which leads to a significant increase in its RMSE. In comparison, the other two adaptive filtering algorithms are more robust since adaptive filtering can adjust process noise adaptively in response to target maneuvers, while the IMM relies on the completeness of its model set. When unknown maneuver patterns appear in target motion, performance degradation caused by model mismatch is likely to occur. Compared with VBAKF, CPAKF has better estimation performance in maneuver segments because it can adjust the noise parameter when a change point is detected. The performance of the three smoothing algorithms, i.e., IMMS, VBAKS, and CPAKS, is superior to the three filtering algorithms. This is because the smoothing algorithms can utilize more measurement information. Nevertheless, their performance deteriorates after a change point occurs. Specifically, CPAKS exhibits better estimation accuracy. This is because CPAKS can divide smoothing intervals through maneuver detection, effectively avoiding the impact on smoothing estimation caused by the state in the interval containing different motion modes of the target. In addition, when a change point is detected, both CPAKF and CPAKS speed up the convergence of parameter estimation by taking decisive action, such as identifying and initializing parameters to avoid falling into the local optimal solution.

The ARMSE of the simulated scenarios is summarized in Table 1. Across all simulated scenarios, CPAKF significantly outperforms the other filtering algorithms. The proposed CPAKS exhibits better smoothing performance compared to IMMS and VBAKS across all scenarios, except for S2. Specifically, CPAKS achieves more pronounced results compared to the other algorithms in the S5 and S6 scenarios because the target maneuvers in these scenarios include more complex forms and higher maneuver frequencies, and CPAKS can better “capture” these prominent change points and provide adaptive interval smoothing state estimation.

The time cost of all simulated scenarios is summarized in Table 2. VBAKF and VBAKS achieve the lowest computational cost compared to the other filtering or smoothing algorithms across all scenarios, followed by CPAKF and CPAKS. IMM and IMMS incur the highest computational burden. Despite the need to estimate an additional discrete variable (run length $r_{k}$) that grows over time for maneuver detection in our algorithm, the accuracy of variational approximation and the fast convergence due to change-point detection results in only a slight increase in computational cost compared to VBAKF and VBAKS.

Among all the algorithms, VBAKF and CPAKF belong to the framework of variational adaptive filtering. In Table 3, we calculate their average iterations in different scenarios. Both algorithms have maximum iteration numbers less than $I_{max}=50$, which indicates good convergence properties. Specifically, CPAKF achieves faster convergence.

Finally, in order to measure the validity and accuracy of change-point detection for CPAKF and CPAKS, $F_{1}$ score metrics were computed, and the results are shown in Table 4. From the results, it can be seen that CPAKF and CPAKS both exhibit capabilities in detecting change points, but CPAKS outperforms CPAKF in change-point detection since CPAKS contains forward filtering and backward smoothing, which leads to better change-point detection performance.

## 6. Results for Real-World Data

To further validate the effectiveness of our methods, real-world data of aerial target tracking were used. The aerial target trajectory is shown in Figure 4, which consists of six different segments or maneuver modes. These segments include segment A (CV), segment B (left CT), segment C (CV), segment D (right CT), segment E (CA), and segment F (figure-eight flight pattern). Through calculation and analysis of the segments, the set of true change points is {13,34,65,81,100} (second).

The parameter settings for all methods in the real-world scenario are similar to those in the simulated scenario. The results are depicted in Figure 5. The ARMSE and average time cost for all methods are summarized in Table 5. The results show that the performance differences of all algorithms in segments A, B, C, and D are not significant. However, the RMSE peak values of CPAKF in segments E and F and CPAKS in segments D, E, and F are significantly smaller than those of the other algorithms, which indicates their better performance in the final ARMSE results. The set of detected change points for CPAKF is {51,75,101} (second), and for CPAKS it is {51,74,100} (second). For M=10, the $F_{1}$ score metric of both CPAKF and CPAKS is 0.5.

## 7. Conclusions

In this article, novel variational adaptive state estimators for target state and process noise parameter estimation are developed for a class of linear state-space models with abruptly changing parameters. By combining variational inference with change-point detection in an online Bayesian fashion, two adaptive estimators—a change-point-based adaptive Kalman filter (CPAKF) and a change-point-based adaptive Kalman smoother (CPAKS)—are proposed in a recursive detection and estimation process. In each step, the run-length probability of the current maneuver mode is first calculated, and then the joint posterior of the target state and process noise parameter conditioned on the run length is approximated by variational inference. Compared with existing variational noise-adaptive Kalman filters, the proposed methods are robust to initial iterative value settings, improving their ability to detect sharply maneuvering targets. Meanwhile, the change-point detection divides the non-stationary time sequence into several stationary segments, allowing for an adaptive sliding length in the CPAKS method. Finally, the performance of the proposed methods is validated using both synthetic and real-world datasets of maneuvering-target tracking.

## Figures and Tables

**Figure 1 sensors-24-04585-f001:**
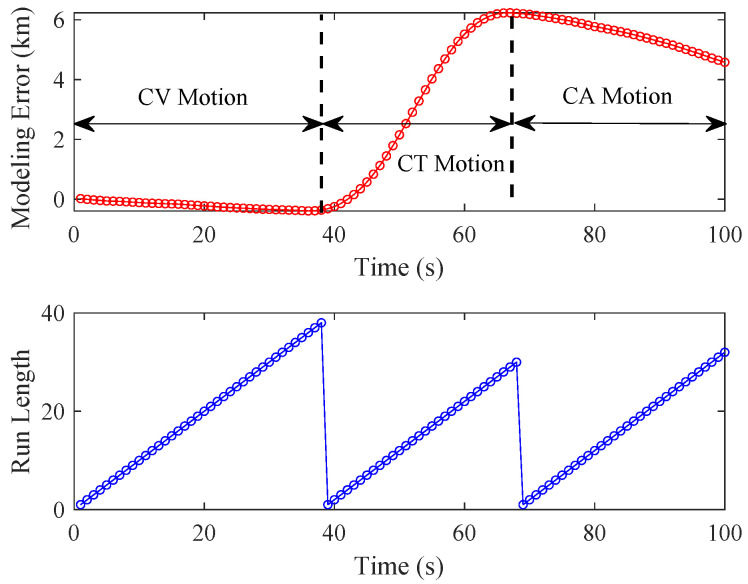
Illustration of maneuvering-target tracking.

**Figure 2 sensors-24-04585-f002:**
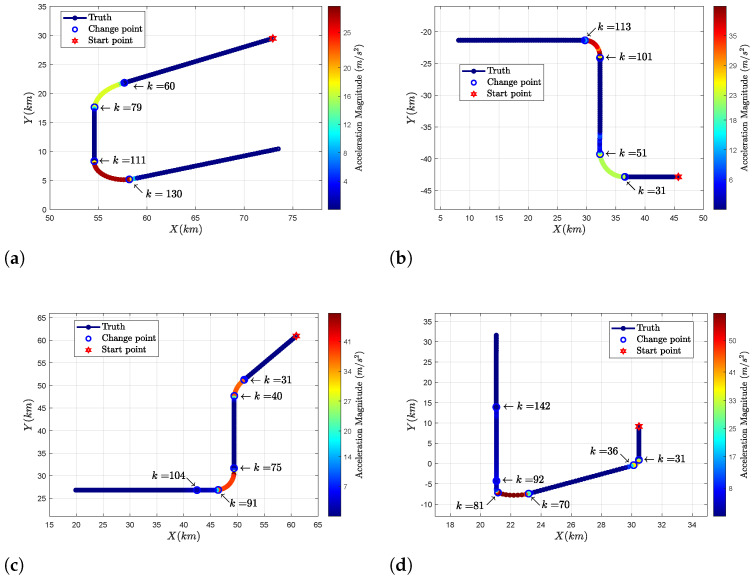
Subfigure (**a**–**f**) are the true trajectories for six simulated scenarios S1–S6.

**Figure 3 sensors-24-04585-f003:**
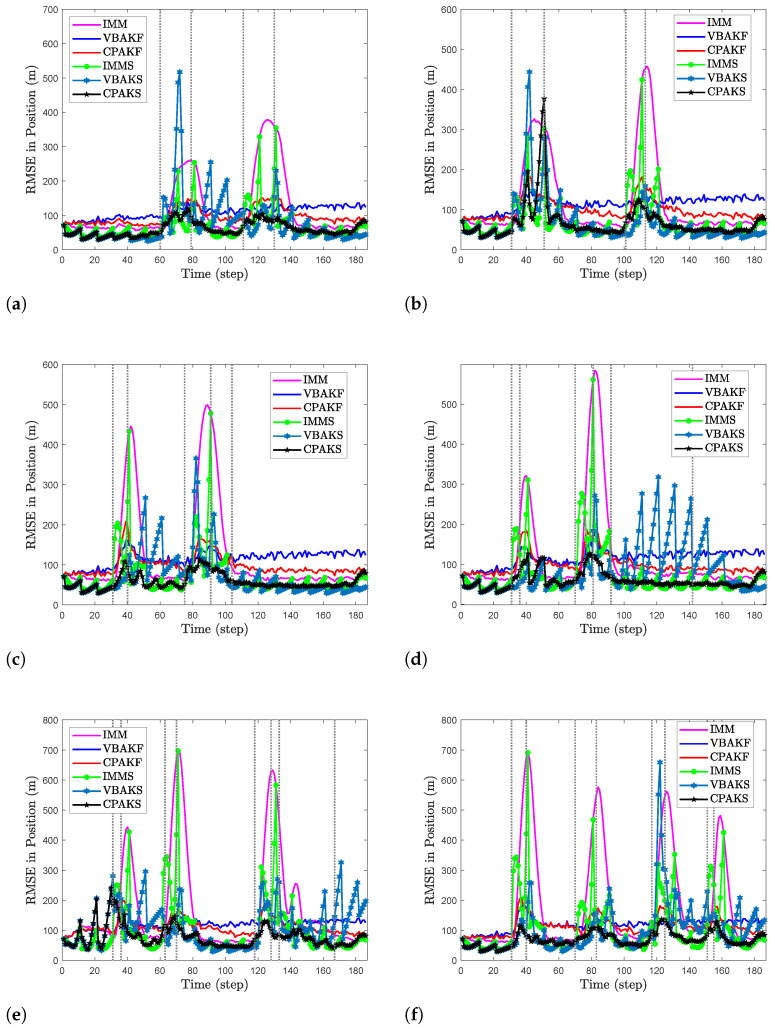
Subfigure (**a**–**f**) exhibit the comparison results of RSME for six simulated scenarios S1–S6.

**Figure 4 sensors-24-04585-f004:**
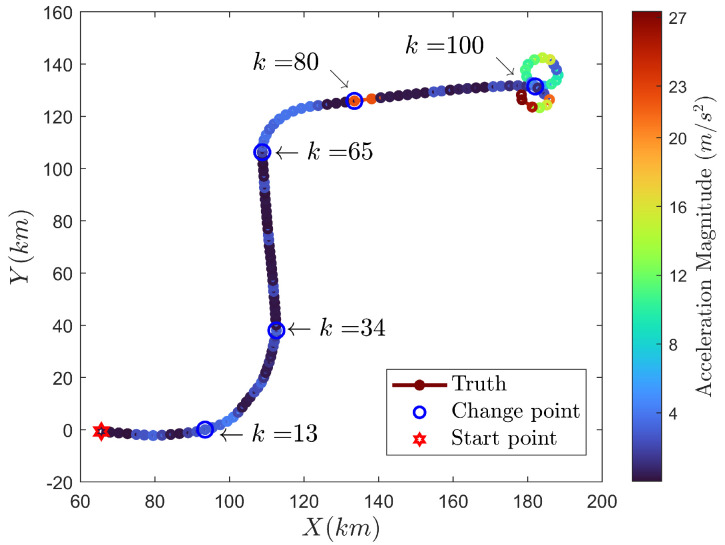
Trajectory for real-world scenario.

**Figure 5 sensors-24-04585-f005:**
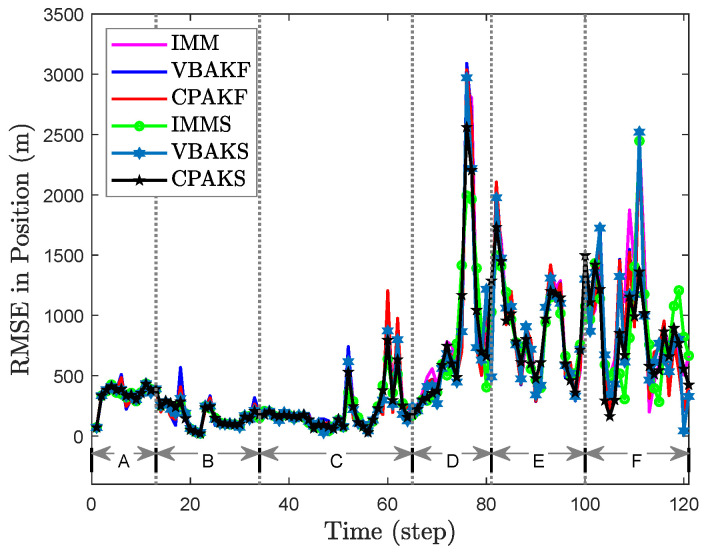
RMSE for real-world scenario.

**Table 1 sensors-24-04585-t001:** ARMSE for all simulated scenarios.

Method	S1	S2	S3	S4	S5	S6
**IMM**	115.55	119.41	120.65	114.32	171.13	184.86
**VBAKF**	109.72	114.70	115.82	116.34	121.19	119.19
**CPAKF**	**97.03**	**100.08**	**98.91**	**99.35**	**111.32**	**113.92**
**IMMS**	68.06	70.62	71.11	73.92	100.25	108.14
**VBAKS**	76.02	**66.66**	70.13	86.59	112.14	102.24
**CPAKS**	**60.32**	67.27	**57.44**	**60.17**	**78.61**	**69.77**

**Table 2 sensors-24-04585-t002:** Time cost for simulated scenarios.

Method	S1	S2	S3	S4	S5	S6
**IMM**	0.80	0.83	0.81	0.79	0.79	0.79
**VBAKF**	0.04	0.04	0.04	0.04	0.04	0.04
**CPAKF**	0.45	0.57	0.46	0.46	0.46	0.46
**IMMS**	1.00	1.02	1.00	0.98	0.98	0.98
**VBAKS**	0.30	0.31	0.31	0.30	0.30	0.30
**CPAKS**	0.80	0.95	0.80	0.80	0.82	0.81

**Table 3 sensors-24-04585-t003:** Average iteration numbers for simulated scenarios.

Method	S1	S2	S3	S4	S5	S6
**VBAKF**	27.57	28.31	26.80	29.41	26.14	27.87
**CPAKF**	9.67	10.08	10.10	10.39	10.94	11.76

**Table 4 sensors-24-04585-t004:** Change-point detection performance.

Method	S1	S2	S3	S4	S5	S6
**CPAKF (M = 5)**	0.25	0.67	0.40	0.60	0.43	0.80
**CPAKS (M = 5)**	0.25	0.67	0.40	0.60	0.43	0.57
**CPAKF (M = 10)**	0.50	0.89	0.60	0.60	0.71	0.93
**CPAKS (M = 10)**	0.50	0.89	0.60	0.60	0.71	0.57

**Table 5 sensors-24-04585-t005:** ARMSE and time cost for real-world scenario.

Method	ARMSE	Time Cost
**IMM**	556.48	0.58
**VBAKF**	537.28	0.05
**CPAKF**	**530.77**	0.24
**IMMS**	510.14	0.69
**VBAKS**	518.50	0.21
**CPAKS**	**507.08**	0.46

## Data Availability

The benchmark trajectories data used in this paper is available at: https://github.com/xlhou/CPAKF-CPAKS (accessed on 4 June 2024).

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
