# Peer review of "Noise-Adaptive State Estimators with Change-Point Detection"

_sensors, 2024, doi:10.3390/s24144585_

Round 1

Reviewer 1 Report

Comments and Suggestions for Authors

The two novel adaptive estimators, variational Bayesian adaptive Kalman filter (VBAKF) changing parameter (CP) and variational Bayesian adaptive Kalman smoother (VBAKS) changing parameter (CP), algorithm, for tracking sharply maneuvering targets is proposed in this manuscript.

The manuscript contains all the necessary elements and follows modern technological challenges in the subject area. However, there are problems and ambiguities in the manuscript that can confuse the reader. The suggestion to the authors is to change the name of the proposed algorithm (and the title of the manuscript) to a shorter and more appropriate one. It is strongly suggested that the authors arrange the terms and abbreviations they use in order to see more clearly what they refer to. Also, each abbreviation must be written in full in the place where it first appears (for example VBAKF-CP, etc…).

In addition to formal and typographical errors, there are also substantial problems in manuscript. The authors chose the method of "adaptation" of the existing algorithm without the necessary evidence that this adaptation is theoretically correct, that is, that the adapted algorithm has the property of convergence after a certain number of iterations. In particular, for the academic public, the dilemma of contribution and the justification of applying this way of solving problems remain. Definition 2.1 (Page 3) is cumbersome, it should not be at the beginning of the Problem Description (Chapter 2). This chapter should contain theoretical models of measurement, noise and target. Definition 1 does not give the reader the necessary theoretical convictions for the statements made. By the way, Remarks 2.1 on the Definition 2.1 also do not say what is its theoretical meaning?

Problems also exist in experiments. A small number of consumption parameters are given, so it is not possible to repeat the experiments without additional parameters. For example, what dynamic models will they use to test the IMM algorithm?

Comments on the Quality of English Language

 Moderate editing of English language required.

Author Response

comment1: The suggestion to the authors is to change the name of the proposed algorithm (and the title of the manuscript) to a shorter and more appropriate one. It is strongly suggested that the authors arrange the terms and abbreviations they use in order to see more clearly what they refer to. Also, each abbreviation must be written in full in the place where it first appears (for example VBAKF-CP, etc…).

Response1: We thank the reviewer for this comment. To address this comment, we have revised the title as “Noise adaptive state estimators with change point detection”. The name of the proposed algorithms have been revised from “VBAKF-CP” and “VBAKS-CP” to “CPAKF” and “CPAKS”. The abbreviation problems have been carefully checked and modified in the revised manuscript. Notations of terms and the corresponding abbreviations has been added on Page 3, Line 125-128.

Comment2: In addition to formal and typographical errors, there are also substantial problems in manuscript. The authors chose the method of "adaptation" of the existing algorithm without the necessary evidence that this adaptation is theoretically correct, that is, that the adapted algorithm has the property of convergence after a certain number of iterations. In particular, for the academic public, the dilemma of contribution and the justification of applying this way of solving problems remain.

Response2: We thank the reviewer for this comment. In the revised manuscript, we have added a remark (\textbf{Remark 3.5.}) to explain the adaptation and the convergence property. Please see the blue-colored text on Page 10, Line 282-289. We copy it in the following for the reviewer's convenience.

\textbf{Remark 3.5.} The proposed CPAKF belongs to  a kind of adaptive kalman filter. This algorithm is suitable for sharply maneuvering target tracking scenarios with unknown noise statistics. The adaptation means that the algorithm can perform noise parameter identification and state estimation simultaneously. The convergence property of the algorithm can be guaranteed in variational inference framework. Some similar proof process is witnessed in [18,26]. According to the coordinate ascent method, we can obtain that the iterative update value of the ELBO is non-decreasing as the fixed-point iteration proceeds. As a result, the iterative update sequence will converge to a local optimum.

Comment3: Definition 2.1 (Page 3) is cumbersome, it should not be at the beginning of the Problem Description (Chapter 2). This chapter should contain theoretical models of measurement, noise and target. Definition 2.1 does not give the reader the necessary theoretical convictions for the statements made. By the way, Remarks 2.1 on the Definition 2.1 also do not say what is its theoretical meaning?

Response3: We thank the reviewer for this comment. To address this comment, we adjust the construct of the Problem Description (Chapter 2) in the revised manuscript. Please see the Page 4, Line 130-168. The theoretical models of measurement, noise and target have been put in the beginning of Chapter 2. We have added a new remark (\textbf{Remark 2.1}) to explain the relationship between sharply maneuvering target and run length after \textbf{Definition 2.1}. The previous remark (\textbf{Remark 2.1.}) has been rewritten as (\textbf{Remark 3.4.}) on page 10, Line 276-281 for illustrating the superiority of our algorithm. We copy it in the following for the reviewer's convenience.

\textbf{Definition 2.1.}[31] Define the discrete random variable $\bm{r}_k \in \{1, \ldots, k\}$ as the run length at time $k$. At each time $k$, the run length $\bm{r}_k$ has only two outcomes: either continues to grow $\bm{r}_k = \bm{r}_{k-1} + 1$ if no change point occurs or drops to one $\bm{r}_k = 1$ when a change point occurs.

\textbf{Remark 2.1.} The run length variable $\bm{r}_k$ is used for describing the change of kinematic model or process noise level. Take Fig.1 as an example for explanation, during the CV motion segment of 0 - 39 \text{s}, the run length continues to grow $\bm{r}_k = \bm{r}_{k-1} + 1$ with $\bm{r}_1 = 1$. At time $ k =40$, the CV motion becomes CT motion, the run length variable at $k =40$ is $\bm{r}_{40} = 1$, and the run length variable begins to grow again $\bm{r}_k = \bm{r}_{k-1} + 1$ with the initial value $\bm{r}_{40} = 1$. Similarly, the run length variable begins to grow again $\bm{r}_k = \bm{r}_{k-1} + 1$ with the initial value $\bm{r}_{70} = 1$ during the CA motion segment.

\textbf{Remark 3.4.}
The CPAKF is distinct from other adaptive estimators. For instance, the IMM estimator assumes several discrete process noise levels and uses a switching rule, whereas CPAKF involves continuous process noise level adjustment through joint estimation. Compared with existing VBAKF [18], CPAKF improves the VB initialization process by incorporating change point detection. Unlike traditional maneuver detection methods, CPAKF relies on Bayesian online change point detection.

[31] R. P. Adams and D. J. MacKay, “Bayesian online changepoint detection,” arXiv preprint arXiv:0710.3742, 2007.

Comment4: Problems also exist in experiments. A small number of consumption parameters are given, so it is not possible to repeat the experiments without additional parameters. For example, what dynamic models will they use to test the IMM algorithm?

Response4: We thank the reviewer for the comment. To address this comment, the parameters of all algorithms have been checked and written in the revised manuscript. The dynamic model of IMM algorithm is assumed to be the constant-velocity model. And the process noise level have been given. Meanwhile, the simulated scenarios are derived from in [35]. The simulated data can be found in Benchmark Trajectories for Multi-Object Tracking tool in Matlab.

Reviewer 2 Report

Comments and Suggestions for Authors

The authors propose an algorithm for tracking sharply maneuvering targets, based on a combination of several known techniques. The topic has been particularly topical in recent years with the mass entry of drones and the need to detect and track them. The subject area is described very well. The algorithm is justified. Several simulation scenarios were used for comparative analysis with some of the currently best algorithms for tracking sharply maneuvering targets. The results show better performance of the newly proposed algorithm. Real target data is also used to compare the same algorithms and again the newly proposed algorithm performs well. The graphic material is of very good quality and adequately illustrates the experiments carried out. All this gives me grounds for a high evaluation of the article proposed for publication.

Some inaccuracies and errors were made in the article, the removal of which will not be difficult and will increase the quality of the presentation of the material even more.

Regarding the theoretical exposition, my only remark is about the justification of the two algorithms for reducing computation complexity (p. 9), which does not sound convincing.

The remaining remarks are rather editorial.

For example, some introduced abbreviations are often used in two different ways (PNC and NPC), other abbreviations are used without prior description or with a subsequent late description (VBAKF-CP - on page 9). The parameters of the InverseWishart distribution are introduced without any description. Overall, despite the relatively large size of the paper, the variational distribution approximation part needs a more detailed description to help the reader. This part is overloaded with mathematical exposition and there is almost no explanatory note. The last two notes are about syntactic errors: "simplicity, We choose" and "the change3point, the".

Author Response

Comment1: Regarding the theoretical exposition, my only remark is about the justification of the two algorithms for reducing computation complexity (p. 9), which does not sound convincing.

Response1: We thank the reviewer for the comment. To address this comment, we have added a reference [31] in the revised manuscript. The work of reducing computation complexity is derived from [31] and has been extensively cited by researchers.

[31] R. P. Adams and D. J. MacKay, “Bayesian online changepoint detection,” arXiv preprint arXiv:0710.3742, 2007.

Comment2: The remaining remarks are rather editorial.
For example, some introduced abbreviations are often used in two different ways (PNC and NPC), other abbreviations are used without prior description or with a subsequent late description (VBAKF-CP - on page 9).

Response2: We thank the reviewer for this comment. To address this comment, We has checked and modified the terms and its corresponding abbreviations, ensuring each abbreviation being written in full in the place where it first appears. For example, the full name of “CPAKF” has been written on Page 3, Line 109-110. Meanwhile, to facilitate the reader's understanding of the correspondence between abbreviations and their full names, we give a notations on Page 3, Line 125-128.

Comment3: The parameters of the Inverse Wishart distribution are introduced without any description.

Response3: We thank the reviewer for this significant suggestion. In the revised manuscript, The description of the Gaussian distribution and Inverse Wishart distribution have been given on Page 5-6, Line 192-198. We copy it in the following for the reviewer's convenience.

The notations $\mathcal{N}(\bm{z}|\bm{d},\bm{D})$ and $\mathrm{IW}(\bm{Z}|a,\bm{A})$ denotes the Gaussian and IW distributions with the PDF given by
\setcounter{equation}{5}
\begin{equation}
    \begin{split}
          \mathcal{N}(\bm{z}|\bm{d},\bm{D}) = \frac{\exp\left[-0.5(\bm{z}-\bm{d})^{\top}\bm{D}^{-1}(\bm{z}-\bm{d})\right]}{\sqrt{(2\pi)^{n}|\bm{D}|}}
    \end{split}
\end{equation}
\begin{equation}
    \begin{split}
         \mathrm{IW}(\bm{Z}|a,\bm{A}) = &\frac{|\bm{A}|^{0.5a}}{2^{0.5n a} \Gamma_n(0.5a)}|\bm{Z}|^{-0.5(a+n+1)}
        \\
       & \times \exp{[-0.5\mathrm{Tr}(\bm{A}\bm{Z}^{-1})]}
    \end{split}
\end{equation}
where $\bm{z} \in \mathbb{R}^{n \times 1}$ is Gaussian random variable with mean $\bm{d}$ and covariance $\bm{D}$; $\bm{Z} \in \mathbb{R}^{n \times n}$ is IW random variable with $a$ and $\bm{A}$ being the degrees of freedom and positive-definite scale matrix, respectively.

Comment4: Overall, despite the relatively large size of the paper, the variational distribution approximation part needs a more detailed description to help the reader. This part is overloaded with mathematical exposition and there is almost no explanatory note.

Response4: We thank the reviewer for this thoughtful suggestion. To describe the derivation process more clearly, we have added a remark (\textbf{Remark 3.2.}) on Page 8, Line 243-248  in the revised manuscript. Some detailed derivations have been given to help the reader on Page 8, Line 251-254.

Comment5: The last two notes are about syntactic errors: "simplicity, We choose" and "the change3point, the".

Response5: We thank the reviewer for the careful comment. In the revised manuscript, we have modified the two syntactic errors on Page 7, Line 222, and Page 10, Line 311.

Round 2

Reviewer 1 Report

Comments and Suggestions for Authors

The two novel adaptive estimators, variational Bayesian adaptive Kalman filter (VBAKF) changing parameter (CP) and variational Bayesian adaptive Kalman smoother (VBAKS) changing parameter (CP), algorithm, for tracking sharply maneuvering targets is proposed in this manuscript.

The new version of the manuscript contains some useful explanations that somewhat give the reader a better insight into the research that led to the proposed algorithm. However, there remain many unclear dilemmas and bold claims without proof of their correctness. It is not easy for the reader to follow the manuscript due to the large number of concepts and equations taken from the literature.

There are typographical errors in the manuscript (eg Page 14, matrix Rk, it says '1e4' instead of ‘10^{4}’ ). The manuscript can still be significant as an idea, but it is necessary to present theoretical evidence in further research (eg convergence of the proposed algorithm).

Comments on the Quality of English Language

Moderate editing of English language required.

Author Response

Response to Reviewer 1

Comment #1: There are typographical errors in the manuscript (eg Page 14, matrix

Rk, it says ’1e4’ instead of ‘104’ ).

Response#1: We thank the reviewer for pointing out the typographical error. in the revised manuscript, we have revised this typographical error on Page 14, Line 371.

Comment #2: The manuscript can still be significant as an idea, but it is necessary to present theoretical evidence in further research (eg convergence of the proposed algorithm).

Response#2: We thank the reviewer for this comment. We agree that the convergence of the proposed algorithm is significant and indispensable. To address this comment, we has added some more detailed analysis and an appropriate reference [34] to explain the convergence of the proposed CPAKF and CPAKS. please see the blue-colored text on Page 10, Line 286-298, and Page 13, Line 355-360. We copy it in the following for the reviewer’s convenience.

Remark 3.6. The convergence of the proposed CPAKF can be explained as

follows. The state estimation and process noise identification of the maneuvering

target tracking is formulated into a Variational optimization problem. Three steps

in CPAKF are implemented to seek for the solutions of this optimization problem.

Firstly, the posterior of run length p(rk|y1:k) can be calculated as (20) via the Bayesian

theorem. Secondly, the conditional posteriors q(xk|rk) and q(Qk|rk) are calculated

as (29) and (35) by maximizing the ELBO (26). Finally, the state posterior q(xk) can

be obtained as (31) via the Bayesian theorem. Since the solutions obtained by the

Bayesian theorem are optimal, the convergence of CVIAKF depends on the procedure

of maximizing the ELBO (26). Recall that maximizing the ELBO is the core in mean

field variational inference [17]. Through the theoretical analysis in [34], one can know

that the mean field variational inference has good convergence guarantees with a

linear convergence rate. Therefore, the proposed CPAKF will gradually converge to

a local optimum as the number of iterations increases.

Remark 4.1. The CPAKS is proposed to improve the estimate accuracy by em- bedding the varational interval smoothing algorithm [33] into the CPAKF. In smooth- ing procedure, the optimization solution of CPAKS are derived from (41) and (42) via the coordinate ascent method. The coordinate ascent method belongs to the fixed iteration method in mean field variational inference [17].According to the convergence 2 guarantee in [34], one can know that the CPAKS can converge to a local optimum as

the number of iterations increases.

[17] D. M. Blei, A. Kucukelbir, and J. D. McAuliffe, “Variational inference: A

review for statisticians,” Journal of the American Statistical Association, vol. 112,

  1. 518, pp. 859–877, 2017.

[34] Zhang A Y and Zhou H H, “Theoretical and computational guarantees of

mean field variational inference for community detection,” The Annals of Statistics,

vol. 48, no. 5, pp. 2575-2598, 2020.

Comment #3: Moderate editing of English language is required.

Response#3: We thank the reviewer for the comment. In the revised manuscript, We have thoroughly reviewed and polished the previous manuscript to correct any grammatical,spelling, and punctuation errors, and to improve overall language fluency and clarity. We also have the paper gone through professional English editing service.
